# Synaptotagmin 7 functions as a Ca²⁺-sensor for synaptic vesicle replenishment

**Huisheng Liu†\*, Hua Bai, Enfu Hui, Lu Yang, Chantell S Evans, Zhao Wang, Sung E Kwon, Edwin R Chapman\***

Department of Neuroscience, Howard Hughes Medical Institute, University of Wisconsin–Madison, Madison, United States

**Abstract** Synaptotagmin (syt) 7 is one of three syt isoforms found in all metazoans; it is ubiquitously expressed, yet its function in neurons remains obscure. Here, we resolved Ca²⁺-dependent and Ca²⁺-independent synaptic vesicle (SV) replenishment pathways, and found that syt 7 plays a selective and critical role in the Ca²⁺-dependent pathway. Mutations that disrupt Ca²⁺-binding to syt 7 abolish this function, suggesting that syt 7 functions as a Ca²⁺-sensor for replenishment. The Ca²⁺-binding protein calmodulin (CaM) has also been implicated in SV replenishment, and we found that loss of syt 7 was phenocopied by a CaM antagonist. Moreover, we discovered that syt 7 binds to CaM in a highly specific and Ca²⁺-dependent manner; this interaction requires intact Ca²⁺-binding sites within syt 7. Together, these data indicate that a complex of two conserved Ca²⁺-binding proteins, syt 7 and CaM, serve as a key regulator of SV replenishment in presynaptic nerve terminals.

**\*For correspondence:** liu25@wisc.edu (HL); chapman@wisc.edu (ERC)

†**Present address:** Waisman Center, University of Wisconsin–Madison, Madison, United States

**Competing interests:** The authors declare that no competing interests exist.

## Introduction

Chemical communication at synapses in the central nervous system is subject to short- and long-term changes in strength (*Malenka, 1994*). For example, the replenishment of synaptic vesicles (SV) in the readily releasable pool (RRP), which can be exhausted during high frequency stimulation (HFS), plays a critical role in determining the rate and degree of short-term synaptic depression (*Wang and Kaczmarek, 1998*). It has been reported that Ca²⁺ plays an important role in the replenishment step (*Dittman and Regehr, 1998*; *Wang and Kaczmarek, 1998*; *Hosoi et al., 2007*; *Sakaba, 2008*). Thus, identification of Ca²⁺ sensors for SV replenishment has emerged as a critical issue, and the ubiquitous Ca²⁺-binding protein CaM has been implicated in this process (*Sakaba and Neher, 2001*; *Hosoi et al., 2007*), by interacting with Munc13-1 (*Lipstein et al., 2013*).

Synaptotagmins represent another well-known family of Ca²⁺-binding proteins; some isoforms function as Ca²⁺-sensors for rapid SV exocytosis (*Chapman, 2008*). Of the 17 syt isoforms in the mammalian genome, only syt 1, 4, and 7 appear to be expressed in all metazoans (*Barber et al., 2009*), suggesting that they are involved in key, evolutionarily conserved, membrane trafficking pathways. In neurons, functions for syt 1 and 4 have been established. Syt 1 serves as a Ca²⁺ sensor for fast synchronous SV release (*Geppert et al., 1994*; *Koh and Bellen, 2003*; *Nishiki and Augustine, 2004*; *Liu et al., 2009*), and syt 4 modulates transmission by regulating neurotrophin release (*Dean et al., 2009*, *2012b*) and also regulates exocytosis in the posterior pituitary (*Zhang et al., 2009*). One group reported that *Syt7* knock-out (KO) mice (hereafter termed KO) had normal synaptic transmission (*Maximov et al., 2008*), but in a more recent study from the same group concluded that syt 7 functions as Ca²⁺ sensor for asynchronous release (*Bacaj et al., 2013*). It was proposed that this contradiction was due to the use of KO mice in the former study, as opposed to the more acute knock-down (KD) approach used in the latter study. However, KD of syt 7 in otherwise WT neurons had no effect (*Bacaj et al., 2013*), so the physiological function of syt 7 in the mammalian central nervous system remains an open issue,

**eLife digest** Neurons communicate with one another at junctions called synapses. The arrival of an electrical signal called an action potential at the first neuron triggers the release of chemicals called neurotransmitters into the synapse. These chemicals then diffuse across the gap between the neurons and bind to receptors on the second cell.

The neurotransmitter molecules are stored in the first cell in packages known as vesicles, which release their contents by fusing with the cell membrane. Following a fusion event, neurons must replenish their vesicle stocks to ensure that they are ready for the arrival of the next action potential. This replenishment process is known to involve a calcium-dependent pathway and a calcium-independent pathway.

A protein called calmodulin, that binds calcium ions, has an important role in the first of these pathways. Now, Liu et al. have shown that another protein, synaptotagmin 7, also has a key role in the replenishment of synaptic vesicles, possibly as a sensor for calcium ions. Moreover, Liu et al. found that synaptotagmin 7 and calmodulin bind to each other to form a complex, which suggests that the calcium-dependent replenishment pathway is regulated by this complex.

The synaptotagmins are a family of 17 proteins, three of which are present in all animals. Two of these were known to have roles in synapses, but the role of the third—synaptotagmin 7—had been unclear. In addition to providing a more complete understanding of the replenishment of synaptic vesicles, the work of Liu et al. also supplies the final piece of the jigsaw regarding the role of the synaptotagmins that are present in all animals.

whereas there is evidence that syt 7 regulates asynchronous synaptic transmission at the Zebrafish neuromuscular junction (*Wen et al., 2010*). The apparent discrepancy regarding the physiological function of syt 7 between mice and Zebrafish is likely due to species differences, analogous to the different functions of syt 4 in *Drosophila* vs mice (*Yoshihara et al., 2005*; *Dean et al., 2009*; *Wang and Chapman, 2010*).

Here, we used high frequency stimulation, and other methods, to study synaptic transmission in cultured hippocamal neurons from KO mice and discovered that syt 7, via a highly specific interaction with CaM, functions as a $Ca^{2+}$-sensor that regulates $Ca^{2+}$-dependent SV replenishment.

## Results

### Loss of syt 7 does not affect spontaneous release, or evoked release triggered by single action potentials

We first carried out whole-cell voltage clamp recordings using primary hippocampal neurons obtained from wild-type (WT) and KO mice and found that spontaneous SV release, measured in the presence of TTX, was unaffected in the KO neurons (*Figure 1A–E*). We then carried out paired recordings by stimulating one neuron with a bipolar glass electrode and recording from a second synaptically connected neuron, as previously described (*Liu et al., 2009*), to examine evoked excitatory postsynaptic currents (EPSCs) triggered by single action potentials (AP). The EPSC amplitudes (*Figure 1F–I,K–L*), and the charge transfer kinetics (*Figure 1J*), were also unchanged in the KOs, confirming that syt 7 does not function as $Ca^{2+}$ sensor for asynchronous release in hippocampal neurons (*Maximov et al., 2008*). We also examined the $Ca^{2+}$-sensitivity of single AP-evoked SV release, by recording at different concentrations of extracellular $Ca^{2+}$, and found that it was the same between WT and KO neurons (*Figure 1K–M*).

### KO neurons exhibit enhanced synaptic depression

We next examined short-term synaptic plasticity in WT and KO neurons by examining the paired pulse ratio (PPR), calculated by dividing the second EPSC (EPSC2) by the first EPSC (EPSC1) (50 ms interval; *Figure 2A*). We also measured synaptic depression during trains of HFS (20 Hz/2.5 s, *Figure 3A*). The PPR was the same between WT and KO neurons (*Figure 2B*), but KO neurons displayed faster depression during HFS (*Figure 3B*). Importantly, expression of WT syt 7, but not the 4D/N $Ca^{2+}$-ligand

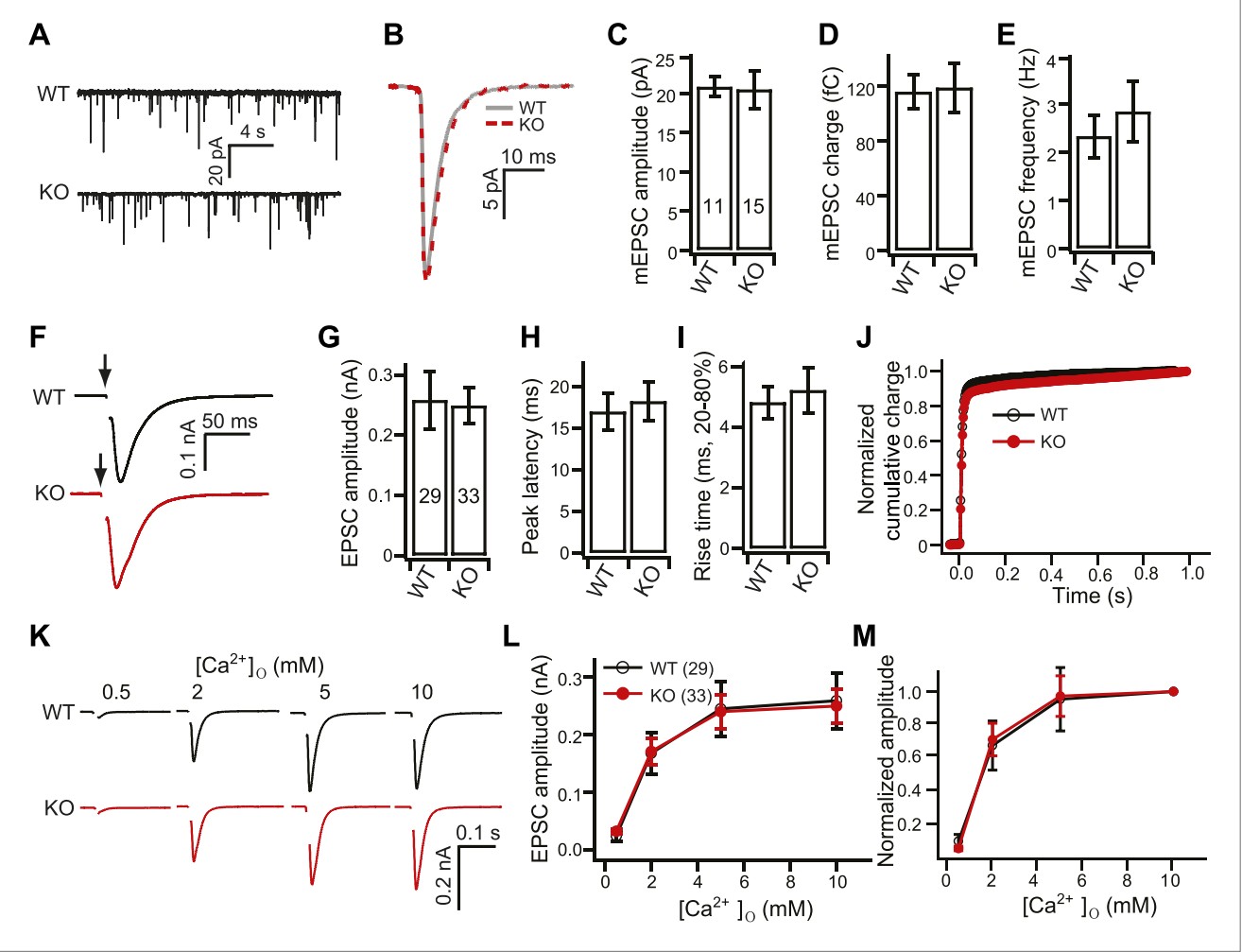

**Figure 1**. Normal spontaneous and single AP evoked EPSCs in KO neurons. (**A**) Representative miniature EPSC (mEPSC; recorded in the presence of TTX) traces from WT and KO hippocampal neurons. (**B**) Representative mEPSCs from WT (grey) and KO (red) neurons. (**C–E**) mEPSC amplitude (**C**, WT: 20.78 ± 1.26 pA; KO: 20.46 ± 2.38 pA), charge (**D**, WT: 116 ± 12 fC; KO: 119 ± 18 fC) and frequency (**E**, WT: 2.32 ± 0.44 Hz; KO: 2.83 ± 0.63 pA) were the same between WT and KO neurons. (**F–J**) Analysis of single AP evoked EPSCs recorded from WT and KO neurons in the presence of 10 mM $[Ca^{2+}]_o$. (**F**) Representative EPSCs recorded from WT and KO neurons. (**G–I**) EPSC amplitude (**G**, WT: 0.25 ± 0.05 nA; KO: 0.24 ± 0.03 nA), peak latency (**H**, WT: 17.03 ± 2.22 ms; KO: 18.27 ± 2.29 ms) and rise time (**I**, WT: 4.8 ± 0.53 ms; KO: 5.21 ± 0.74 ms) were the same between WT and KO neurons. (**J**) The normalized cumulative total charge was fitted with a double exponential function (solid line). WT and KO neurons had identical charge transfer kinetics. (**K**) Representative single AP evoked EPSCs recorded from WT and KO neurons in the presence of the indicated $[Ca^{2+}]_o$. (**L–M**) EPSC amplitudes (**L**) and relative amplitudes (**M**) plotted vs $[Ca^{2+}]_o$. The n values indicate the number of neurons from two independent litters of mice; these values are provided in the bars in all figures in this study. All data represent mean ± SEM. Statistical significance was analyzed using the Student's *t* test.

mutant (which was properly folded but failed to bind $Ca^{2+}$ (*Figure 3—figure supplement 1*)), rescued the depression phenotype (*Figure 3B*). We note that WT syt 7, and the 4D/N mutant, were both present in synapses but were not well colocalized with the SV marker, synapsin, as compared to syt 1 (*Figure 3—figure supplement 2*). The surface fractions, as well as the internal fractions, were similar between syt 1 and 7 measured by using a pHluorin tag (*Figure 3—figure supplement 3*), consistent with a previous report (*Dean et al., 2012a*).

Five mechanisms could potentially contribute to the faster depression observed in KO neurons: changes in postsynaptic receptor density or kinetic properties; saturation or desensitization of postsynaptic receptors; a smaller RRP size which fails to supply sufficient SVs; defective SV replenishment; or slower SV recycling/refilling. Below, we address each of these possibilities and demonstrate that defective SV replenishment underlies the increased rate of depression in KO neurons.

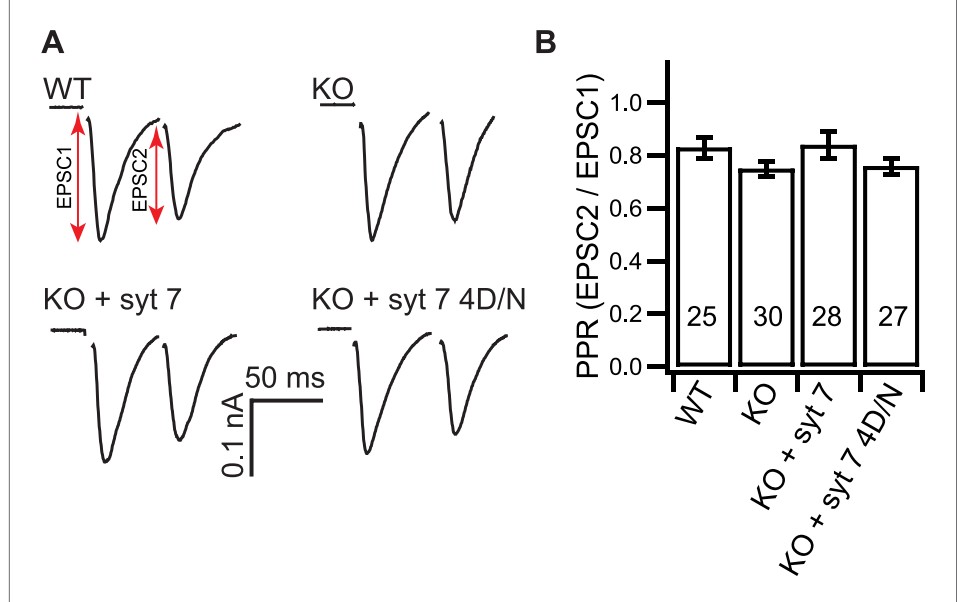

**Figure 2**. Loss of syt 7 does not affect the PPR. (**A**) Representative EPSCs recorded in paired-pulse experiments, with 50 ms inter-stimulus intervals, from WT, KO, or KO neurons expressing either WT (KO + syt 7) or the $Ca^{2+}$ ligand mutant form (KO + syt 7 4D/N) of syt 7. (**B**) The PPRs were the same among all groups (WT: 0.83 ± 0.04; KO: 0.75 ± 0.03; KO + syt 7: 0.84 ± 0.05; and KO + syt 7 4D/N: 0.76 ± 0.03). The n values indicate the number of neurons, obtained from three independent litters of mice, used for these experiments. All data represent mean ± SEM. Statistical significance was analyzed by one-way ANOVA.

## Loss of syt 7 impairs SV replenishment

The unaltered quantal release properties in KO neurons (*Figure 1A–E*) rule out changes in postsynaptic components, such as the density or kinetic properties of AMPA receptors. To further exclude the role of desensitization or saturation of postsynaptic receptors during HFS, we monitored transmission, during HFS, in the presence of cyclothiazide (CTZ, 50 µM) and kynurenic acid (KYN, 100 µM), as described previously (*Neher and Sakaba, 2001*; *Figure 3—figure supplement 4A*). CTZ and KYN reduce AMPA receptor desensitization and saturation, respectively. KO neurons still displayed faster depression (*Figure 3—figure supplement 4B*).

Next, we integrated the phasic charge from the HFS experiments (*Figure 3A*) and plotted the cumulative charge vs time (*Figure 3C*). The last ten EPSCs were best fitted by a linear function (solid line in *Figure 3C*; linearity is established in the inset), as described in previous reports utilizing hippocampal neurons (*Stevens and Williams, 2007*) and the calyx of Held (*Schneggenburger et al., 1999*; *Hosoi et al., 2007*), to calculate RRP size, release probability, and SV replenishment rates. The size of the RRP (the y-intercept) was similar between WT and KO neurons (*Figure 3D*). Since single AP evoked EPSCs were identical between WT and KO neurons (*Figure 1F*), SV release probability (calculated by dividing the EPSC charge by the RRP size) was unchanged in the KOs (0.21 ± 0.03 for KO vs 0.24 ± 0.02 for WT). The slope of the linear function was divided by the y-intercept in a given cell pair to calculate the SV replenishment rate. Strikingly, we found that the replenishment rate was reduced by ~50% for phasic release in KO neurons (0.26 ± 0.03 $s^{-1}$ for KO vs 0.52 ± 0.06 $s^{-1}$ for WT) (*Figure 3E*). WT syt 7, but not the 4D/N mutant, fully rescued SV replenishment (*Figure 3C,E*). The size of the RRP was unchanged in the rescue experiments (*Figure 3D*).

We also applied similar methods as described in *Figure 3C* to investigate SV replenishment during tonic transmission (*Figure 3—figure supplement 5A,B*); the replenishment rate was, again, reduced by ~50% in KO neurons (2.19 ± 0.22 $s^{-1}$ for KO vs 3.95 ± 0.61 $s^{-1}$ for WT), and WT syt 7, but not the 4D/N mutant, rescued this defect (*Figure 3—figure supplement 5C*). These results are consistent with a previous report indicating that tonic transmission is maintained by $Ca^{2+}$-dependent SV replenishment during activity (*Otsu et al., 2004*).

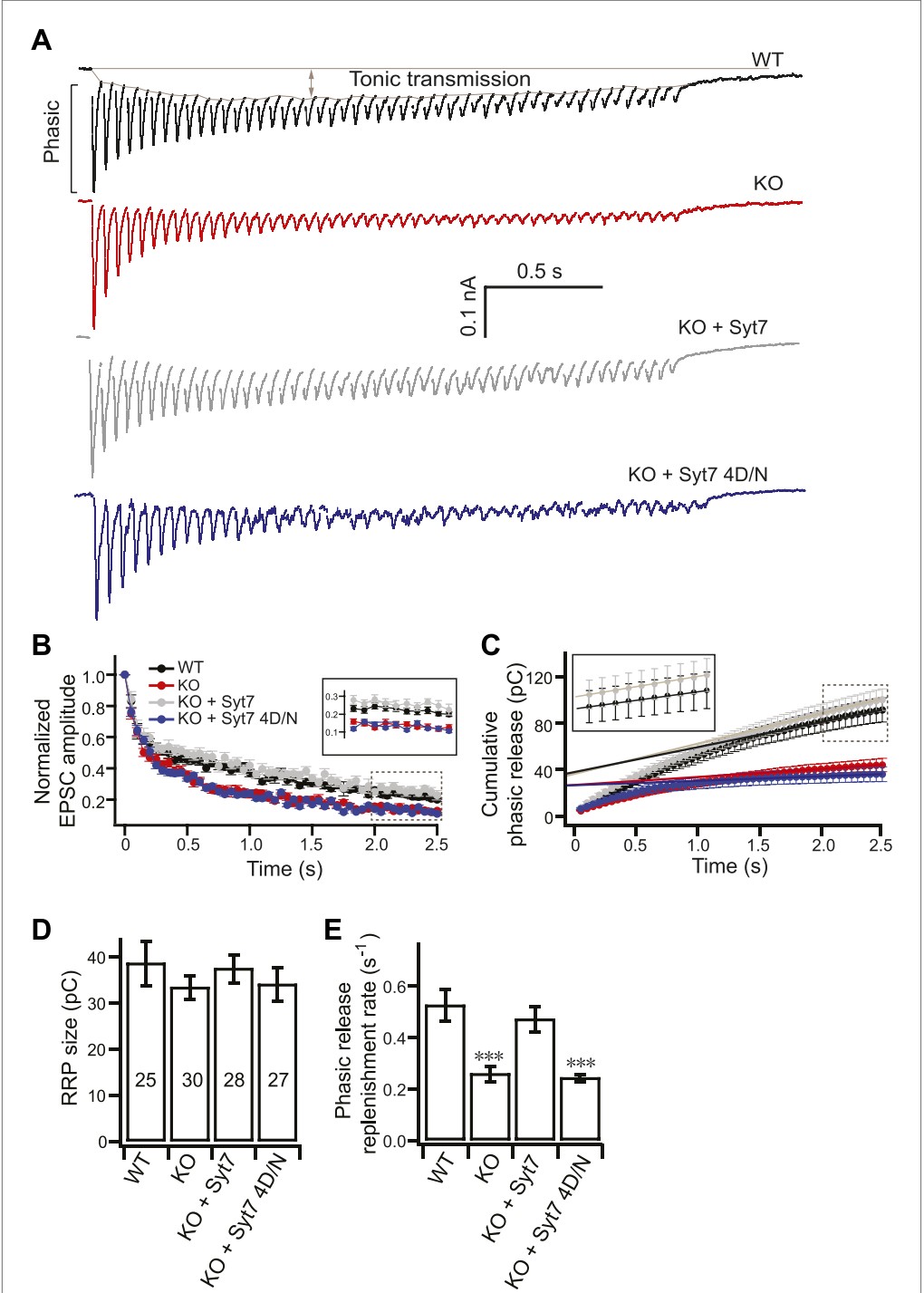

**Figure 3**. SV replenishment is reduced in KO neurons. (**A**) Representative EPSC traces (20 Hz/2.5 s) recorded from WT (black), KO (red), or KO neurons expressing either WT (KO + syt 7, gray) or the 4D/N mutant form of syt 7 (KO + syt 7 4D/N, blue). The phasic and tonic components of transmission are indicated within the representative WT trace. (**B**) The peak amplitude of each EPSC was divided by the peak of the first response and plotted vs time. Inset: expansion of the last 10 responses. (**C**) Plot of cumulative total phasic charge transfer vs time. Data points from the last ten EPSCs were fitted with a linear function to calculate the RRP size (y-intercept in panel **D**), and SV replenishment rate (the slope divided by the RRP size; panel **E**). Inset: expansion of the last ten responses. (**D**) The RRP size is the same among all groups (WT: 38.70 ± 4.78 pC; KO: 33.49 ± 2.48 pC; KO + syt 7: 37.58 ± 3.02 pC; and KO + syt 7 4D/N: 34.16 ± 3.63 pC). (**E**) The SV replenishment rate was reduced in KO ($0.26 \pm 0.03$ s$^{-1}$) as compared to WT

*Figure 3. Continued on next page*

*Figure 3. Continued*

neurons (0.52 ± 0.06 s$^{-1}$). WT (0.47 ± 0.05 s$^{-1}$), but not 4D/N mutant syt 7 (0.24 ± 0.02 s$^{-1}$), rescued SV replenishment. The n values indicate the number of neurons, obtained from three independent litters of mice, used for these experiments. All data shown represent mean ± SEM. \*\*\*p<0.001, analyzed by one-way ANOVA, compared to WT.

The following figure supplements are available for figure 3:

**Figure supplement 1**. Ca$^{2+}$-ligand mutations disrupt Ca$^{2+}$ binding, but not proper folding, of syt 7.

**Figure supplement 2**. Syt 7 is not localized to SVs.

**Figure supplement 3**. Relative distribution of syt 7 in the plasma membrane, vs internal compartments, in hippocampal neurons.

**Figure supplement 4**. Postsynaptic receptor desensitization and saturation do not contribute to the KO phenotype.

**Figure supplement 5**. Replenishment of the tonic component of transmission is reduced in KO neurons.

## Slower RRP recovery in KO neurons

To further confirm the replenishment defect in KO neurons, we investigated the recovery kinetics of phasic release after HFS. Recovery was calculated by measuring the total charge transfer during phasic release at various time intervals, between bouts of HFS, where the second response was divided by the first response (*Figure 4A*). A double exponential function was used to fit the recovery time-course, revealing fast and slow components of recovery. Notably, fast recovery (*Figure 4B*) was delayed by ~60% in KO neurons (1.32 ± 0.22 s) as compared to WT neurons (0.55 ± 0.17 s), while slow

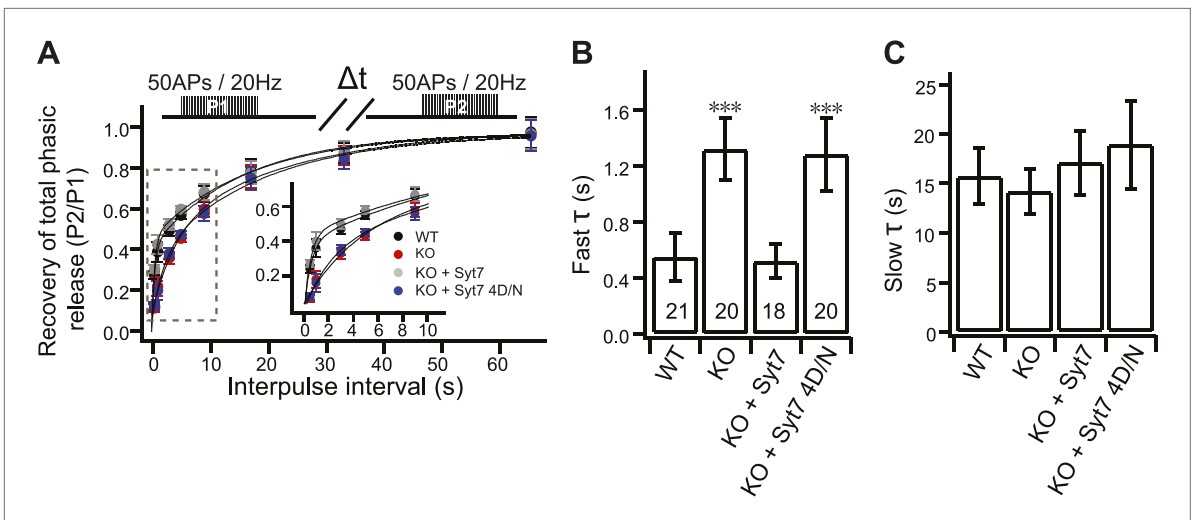

**Figure 4**. Impaired fast recovery of phasic transmission in KO neurons. (**A**) Time course for the recovery of phasic release (Δt = 0.5, 1, 3, 5, 9, 17, 33 and 65 s). Data were fitted using a double exponential function, yielding fast (**B**) and slow (**C**) recovery. The upper diagram is the stimulation protocol. Inset: expansion of the early phase of the plot. (**B**) The fast recovery time constant was increased in KO (1.32 ± 0.22 s) as compared to WT neurons (0.55 ± 0.17 s). WT syt 7 (0.52 ± 0.12 s), but not the 4D/N mutant (1.28 ± 0.26 s), rescued the SV replenishment rate in KO neurons. (**C**) The slow recovery time constant was invariant among all groups. (WT: 15.72 ± 2.85 s; KO: 14.13 ± 2.24 s; KO + syt 7: 17.03 ± 3.24 s; and KO + syt 7 4D/N: 18.88 ± 4.45 s). The n values indicate the number of neurons, obtained from three independent litters of mice, used for these experiments. All data shown represent mean ± SEM. \*\*\*p<0.001, analyzed by one-way ANOVA, compared to WT.

The following figure supplements are available for figure 4:

**Figure supplement 1**. The fast recovery component of tonic transmission is impaired in KO neurons.

recovery (*Figure 4C*) was unaffected by loss of syt 7 (WT: 15.72 ± 2.85 s; KO: 14.13 ± 2.24 s). WT syt 7 (0.52 ± 0.12 s), but not the 4D/N mutant (1.28 ± 0.26 s), rescued fast recovery (*Figure 4A,B*), indicating that syt 7 regulates this process in a $Ca^{2+}$-dependent manner during phasic release. Surprisingly, the recovery of total tonic transmission during HFS also displayed the same two recovery components (*Figure 4—figure supplement 1A–C*). WT syt 7, but not the 4D/N mutant, rescued fast recovery of tonic transmission (*Figure 4—figure supplement 1A,B*). Thus, recovery of phasic and tonic release might involve the same $Ca^{2+}$-dependent pathway.

## Enhanced synaptic depression in KO neurons during 'first fusion' events

To address potential defects in SV recycling/refilling in KO neurons during ongoing transmission, we utilized synaptophysin-pHluorin (sypHy) to measure SV exocytosis driven by HFS (20 Hz) in the presence of bafilomycin (1 µM), an inhibitor of the vesicular proton pump (*Figure 5A–C*), to measure first fusion events. Under these conditions, in which SV recycling cannot play a role in ongoing synaptic transmission, KO neurons still displayed a significant reduction in SV release (*Figure 5B*). Importantly, KO neurons exhibited reductions in SV release as early as the second imaging time point (i.e., 1 sec, 20 Hz stimulation) (*Figure 5C*), consistent with electrophysiological recordings in *Figure 3C*. Moreover, we also examined first release events during HFS by recording synaptic currents in the presence of bafilomycin and, again, significantly faster depression (*Figure 5D*), reduced cumulative release (*Figure 5E*), unchanged RRP size (*Figure 5F*), and slower SV replenishment (*Figure 5G*), were observed in the KOs. These data indicate that SV recycling does not contribute to the observed reduction in SV release in KO neurons during HFS.

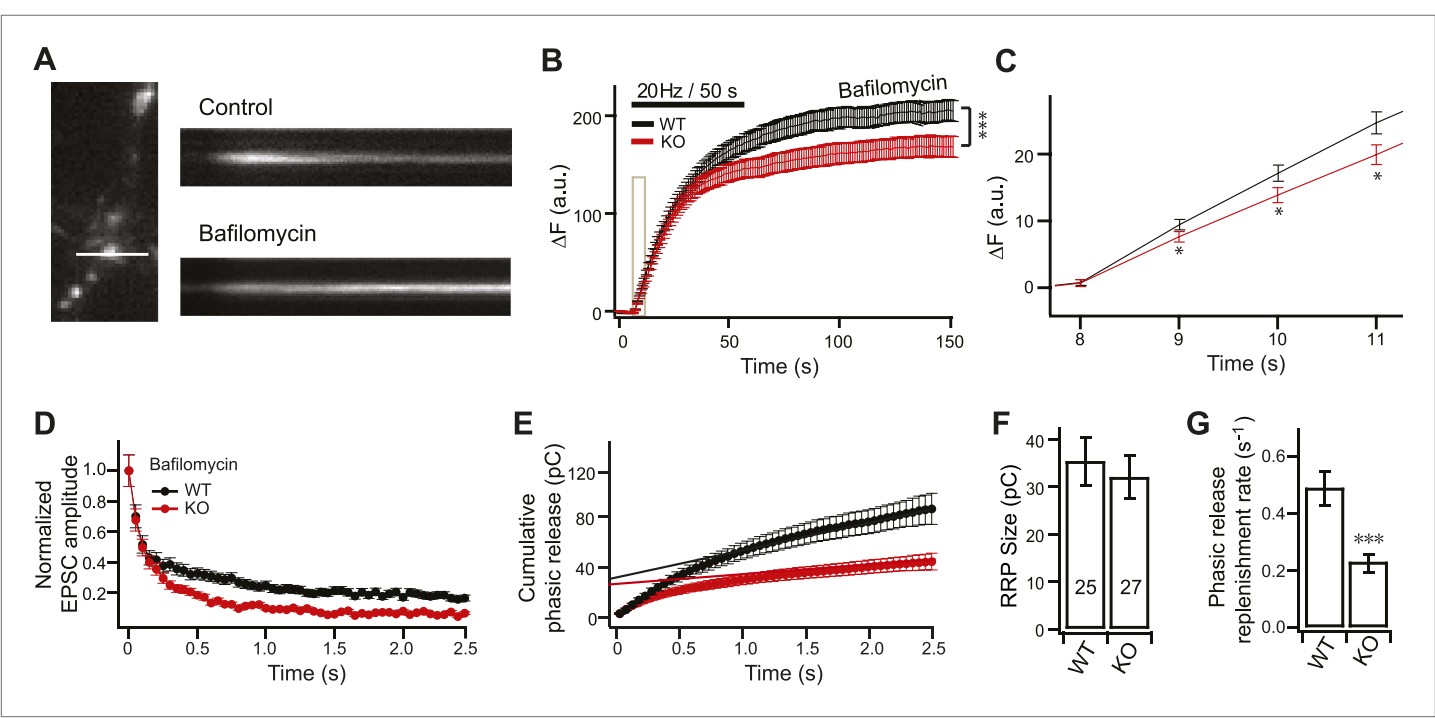

**Figure 5**. The reduction in SV release in KO neurons during HFS is independent of vesicle recycling. (**A**) Representative image showing the sypHy signal from the same synapse upon stimulation in the absence, and presence, of bafilomycin. (**B**) Averaged traces of sypHy in WT (black) and KO (red) neurons in response to stimulation in the presence of bafilomycin. KO neurons displayed a significant reduction in sustained SV release as compared to WT neurons. Data were collected for ~300 boutons per condition as follows: ~50 boutons per coverlip, from six coverslips (two each from three independent litters of mice). All data shown represent mean ±SEM. ***p<0.001 analyzed by two-way ANOVA. The initial response, indicated by the tan rectangle, is expanded in panel **C**. (**C**) KO neurons display reduced sypHy responses early in the train. *p<0.05, analyzed using the Student's *t* test. (**D–G**) ESPC recordings, in response to HFS, in the presence of bafilomycin. (**D**) The peak amplitude of each EPSC was normalized to the peak of the first response and plotted vs time. (**E**) Plot of cumulative total phasic charge transfer vs time. Data were fitted and analyzed as described in *Figure 3C*. (**F**) The size of the RRP is the same in WT and KO neurons (WT: 35.42 ± 5.07 pC; KO: 32.07 ± 4.54 pC). (**G**) Reduction in the SV replenishment rate in KO neurons in the presence of bafilomycin (WT: 0.49 ± 0.06 s⁻¹; KO: 0.23 ± 0.03 s⁻¹). The n values indicate the number of neurons, obtained from three independent litters of mice, used for these experiments. All data shown represent mean ± SEM. ***p<0.001, analyzed using the Student's *t* test.

## Syt 7 functions as a Ca²⁺-sensor for Ca²⁺-dependent SV replenishment

Previous studies demonstrated that $Ca^{2+}$ plays a critical role in the replenishment of SVs (*Dittman and Regehr, 1998*; *Wang and Kaczmarek, 1998*; *Hosoi et al., 2007*; *Sakaba, 2008*). The syt 7 4D/N mutant, which fails to bind $Ca^{2+}$, was unable to rescue SV replenishment in KO neurons (*Figure 3*), suggesting that syt 7 functions as $Ca^{2+}$-sensor during replenishment. To test whether $Ca^{2+}$ is essential for the function of syt 7, we measured recovery of the RRP by applying paired pulses of hypertonic sucrose (500 mM, each for 10 s), with varying interpulse time intervals, under $Ca^{2+}$-free conditions (replacement of $[Ca^{2+}]_o$ with EGTA, incubation with BAPTA-AM, and addition of cyclopazonic acid [CPA] to empty internal stores [*Liu et al., 2009*]) (*Figure 6A*). For comparison, we also measured RRP recovery under physiological $[Ca^{2+}]_o$ (2 mM; *Figure 6A*).

Consistent with the same RRP size measured by HFS among different groups (*Figure 3D*), the RRP size, calculated from the first response to sucrose, was the same between WT, KO, and KOs expressing WT or the syt 7 4D/N mutant, in physiological $[Ca^{2+}]_o$ (*Figure 6B*), excluding a role for syt 7 in SV priming. Moreover, the RRP size was the same between the $Ca^{2+}$-free and physiological $[Ca^{2+}]_o$ conditions (*Figure 6B*). Of note, there was a significant difference in the absolute size of the RRP as measured using hypertonic sucrose (*Figure 6B*; ~8000 SVs, calculated by dividing the RRP charge by the quantal charge) vs HFS (*Figure 3D*; ~300 SVs). This was expected, because during electrical stimulation, only the synapses formed between the recorded neuron and stimulated neuron are measured. In contrast, hypertonic sucrose activates all of the synapses that impinge on the recorded neuron.

We then plotted the ratio of the second (P2) relative to the first (P1) sucrose response, as a function of the interpulse time interval, to calculate the recovery time course of the RRP in physiological $[Ca^{2+}]_o$ (*Figure 6C*) and $Ca^{2+}$-free conditions (*Figure 6D*); these data were best fitted by a single exponential function (solid line in *Figure 6C,D*). In physiological $[Ca^{2+}]_o$, recovery of the RRP was slowed by ~60% in the KOs (18.03 ± 2.87 s for KO vs 7.29 ± 1.05 s for WT) (*Figure 6E*), which is similar to the findings reported above using HFS (*Figure 4B*). WT syt 7 (8.30 ± 0.79 s), but not the 4D/N mutant (20.07 ± 2.63 s), rescued this phenotype (*Figure 6C,E*). Remarkably, in $Ca^{2+}$-free conditions (*Figure 6D*), the recovery of the RRP in WT neurons was reduced to the same level as observed for KO neurons, with or without expression of the 4D/N mutant in physiological $[Ca^{2+}]_o$ conditions (*Figure 6E*). Together, the results thus far suggest that there are $Ca^{2+}$-dependent and $Ca^{2+}$-independent pathways for replenishment, and that syt 7 functions as a $Ca^{2+}$-sensor that regulates the $Ca^{2+}$-dependent pathway.

Hypertonic sucrose stimulates SV release without increasing $[Ca^{2+}]_i$ (*Yao et al., 2011*, *2012*). Therefore, the $Ca^{2+}$-dependent component for recovery of the RRP, observed in the hypertonic sucrose experiments, is likely due to the resting $[Ca^{2+}]_i$. In vitro reconstitution experiments revealed that syt 7 begins to drive fusion at $[Ca^{2+}] > 100$ nM ($[Ca^{2+}]_{1/2} = 300$ nM) (*Bhalla et al., 2005*). Hence, it is plausible that syt 7 can indeed sense resting $[Ca^{2+}]_i$ to regulate $Ca^{2+}$-dependent SV replenishment.

## Syt 7 cooperates with CaM to regulate Ca²⁺-dependent SV replenishment

The reduced SV replenishment phenotype in KO neurons is reminiscent of the effects of CaM antagonists on vesicle replenishment in the calyx of Held (*Sakaba and Neher, 2001*; *Hosoi et al., 2007*), raising the possibility that there are links between CaM and syt 7; indeed, syt 1 has been reported to bind CaM (*Popoli, 1993*; *Perin, 1996*). We therefore conducted pull-down assays using immobilized CaM and the soluble cytoplasmic domains (C2AB) of syt 1, 2, 4, 7, 9 and 10 (*Figure 7A*). Surprisingly, robust $Ca^{2+}$-dependent binding to CaM was observed only for syt 7, demonstrating the specificity of the interaction (note: low levels of binding of syt 1, or other syt isoforms, to CaM might have escaped detection). Moreover, the $Ca^{2+}$-promoted interaction between syt 7 and CaM was abolished by the $Ca^{2+}$-ligand mutations in the syt 7 4D/N mutant (*Figure 7A*) and was specific for $Ca^{2+}$ as $Mg^{2+}$ was without effect (*Figure 7B*). Together, these biochemical findings are consistent with a model in which the $Ca^{2+}$-sensor for replenishment is actually a complex formed between these two conserved $Ca^{2+}$-binding proteins, syt 7 and CaM.

To address the potential contribution of CaM to SV replenishment in hippocampal neurons, we next analyzed the SV replenishment rate in WT and KO neurons in the presence and absence of calmidazolium (CDZ; 20 µM), a cell-permeable CaM antagonist (*Sakaba and Neher, 2001*; *Figure 7C*). Cells were incubated for 30 min prior to recording to ensure that CDZ had taken effect, and then HFS trains were applied. WT neurons displayed faster depression (*Figure 7D*, upper) and reduced cumulative phasic release (*Figure 7D*, lower) in the presence of CDZ, consistent with previous observations based on the calyx of Held (*Sakaba and Neher, 2001*; *Hosoi et al., 2007*). Similarly, cumulative tonic transmission

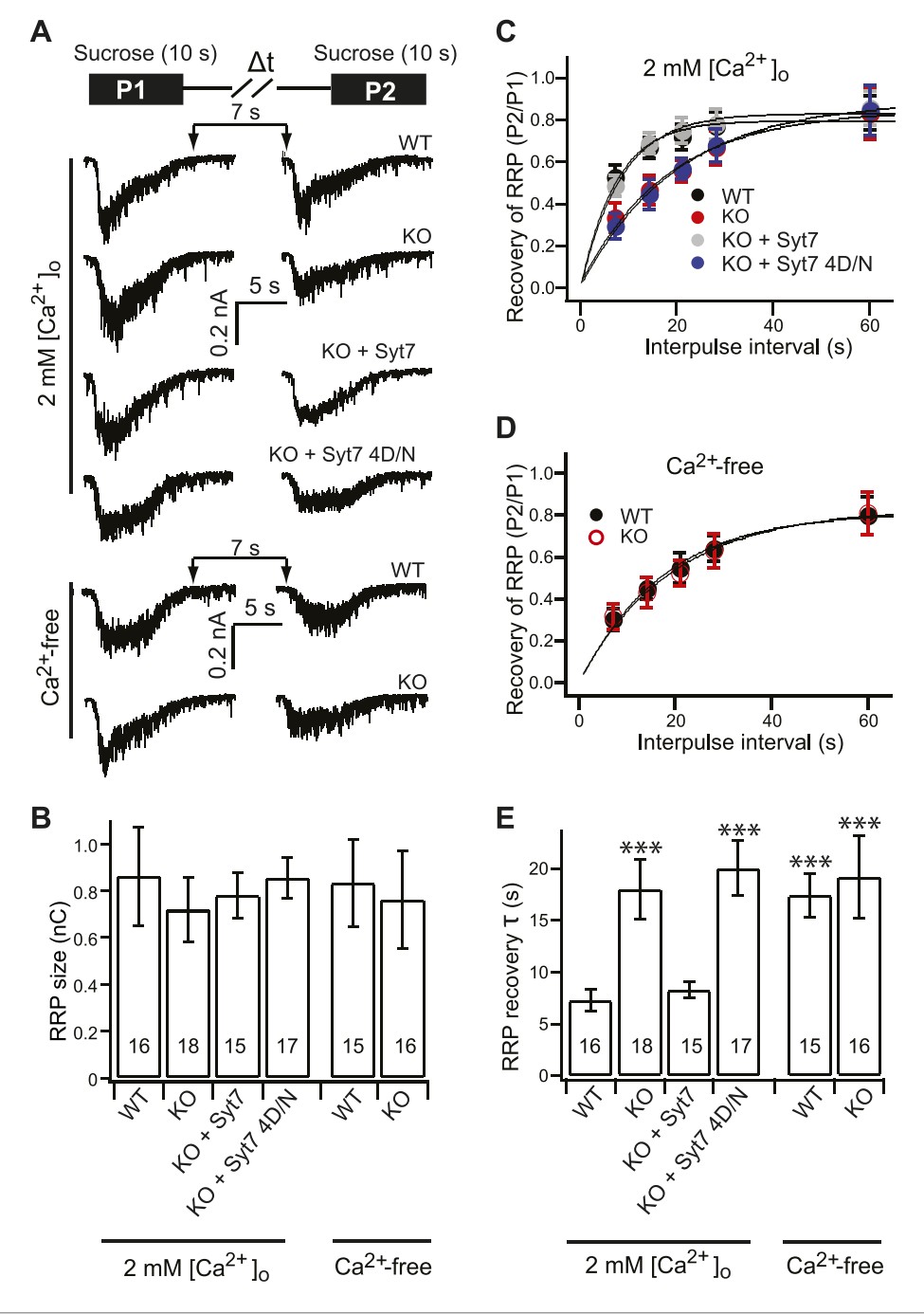

**Figure 6**. Syt 7 functions as a $Ca^{2+}$-sensor for $Ca^{2+}$-dependent SV replenishment. (**A**) Upper panel illustrates the sucrose application protocol ($\Delta t$ = 7, 14, 21, 28 and 60 s). Lower panel shows typical responses to sequential application of sucrose (7 s interval) from different groups in 2 mM $[Ca^{2+}]_o$ or $Ca^{2+}$-free conditions. (**B**) The RRP size, calculated from the first sucrose response, was same among the all groups (in 2 mM $[Ca^{2+}]_o$, WT: 0.86 ± 0.21 nC; KO: 0.72 ± 0.14 nC; KO + syt 7: 0.78 ± 0.09 nC; and KO + syt 7 4D/N: 0.85 ± 0.08 nC; in $Ca^{2+}$-free conditions, WT: 0.83 ± 0.18 nC; KO: 0.76 ± 0.2 nC). (**C**) The time course of RRP recovery in 2 mM $[Ca^{2+}]_o$ conditions between applications of hypertonic sucrose. Data were fitted with a single exponential function to calculate the recovery time constant in panel **E**. (**D**) The recovery of the RRP in $Ca^{2+}$-free conditions. Data were fitted with a single exponential function to calculate the recovery time constant in panel **E**. (**E**) In 2 mM $[Ca^{2+}]_o$, recovery of the RRP was significantly slowed in KO (18.03 ± 2.87 s) neurons as compared to WT (7.29 ± 1.05 s) neurons, and was

*Figure 6. Continued on next page*

*Figure 6. Continued*

rescued by WT syt 7 (8.30 ± 0.79 s), but not the 4D/N mutant (20.07 ± 2.63 s). In $Ca^{2+}$-free conditions, RRP recovery in WT neurons (17.42 ± 2.11 s) was slowed to the same level as in KO neurons (19.2 ± 3.95 s). The n values indicate the number of neurons, obtained from three independent litters of mice, used for these experiments. All data shown represent mean ± SEM. ***p<0.001, analyzed using one-way ANOVA, compared to WT.

in WT neurons was also reduced by CDZ (*Figure 7—figure supplement 1A*). Interestingly, both cumulative phasic and tonic release in WT neurons were reduced to the same levels as in the KO neurons, which were not affected by CDZ (*Figure 7C*, *Figure 7—figure supplement 1A*). These experiments also revealed that: (1) CDZ does not change the size of the RRP (*Figure 7E*), and (2) in WT neurons, the replenishment rates for phasic (*Figure 7F*) and tonic release (*Figure 7—figure supplement 1B*) were both reduced to the same levels observed for the KOs, with or without CDZ.

## A CaM antagonist inhibits recovery of the RRP in WT neurons, phenocopying loss of syt 7

We next examined the recovery of phasic (*Figure 8A*) and tonic release (*Figure 8—figure supplement 1A*), as described in *Figure 4*, in the presence of CDZ during HFS, in WT and KO neurons. The time course for fast recovery, during phasic (*Figure 8B*) and tonic (*Figure 8—figure supplement 1B*) transmission in WT neurons, was impaired, phenocopying the KO neurons which, in turn, were unaffected by CDZ. The time course for slow recovery was unaffected under all conditions tested (*Figure 8C*, *Figure 8—figure supplement 1C*).

We also used hypertonic sucrose, in 2 mM $[Ca^{2+}]_o$ and in the presence of CDZ, to measure the time course for recovery of the RRP in WT and KO neurons (*Figure 8D*). CDZ impaired the RRP recovery kinetics in WT neurons, again phenocopying the KOs, which were not affected by the drug (*Figure 8E,F*).

The findings reported above indicate that CaM and syt 7 might act together to regulate vesicle replenishment. We therefore assayed the effect of CDZ on this interaction. CDZ, up to 100 μM, was simultaneously incubated with the syt 7 C2AB domain and immobilized CaM. Competition was not observed (*Figure 8—figure supplement 2*). We also incubated immobilized CaM with CDZ for 30 min prior to the addition of syt 7 C2AB. Again, CDZ had no effect on syt 7·CaM interactions (data not shown). These findings indicate that syt 7 and CDZ occupy different sites on CaM to regulate SV replenishment. Thus, CaM might serve as an adaptor, linking syt 7 to another CDZ-sensitive effector in the $Ca^{2+}$-dependent replenishment pathway.

## Discussion

Short term synaptic plasticity (STP) plays an important role in the central nervous system, including learning and memory (*Zucker and Regehr, 2002*). Changes in SV release probability, RRP size, SV replenishment, or SV recycling can impact STP. In the current report, we discovered that syt 7 contributes to STP by regulating SV replenishment.

WT syt 7, but not the 4D/N mutant that fails to bind $Ca^{2+}$, rescues the defects in SV replenishment observed in KO neurons, indicating that syt 7 functions as $Ca^{2+}$ sensor for this process. We further distinguished a $Ca^{2+}$-independent SV replenishment pathway from the $Ca^{2+}$-dependent pathway by using hypertonic sucrose to drive secretion in the complete absence of $Ca^{2+}$ (*Figure 6*). $Ca^{2+}$-independent SV replenishment has been discussed in previous studies (*Stevens and Wesseling, 1998*; *Hosoi et al., 2007*); however, only the slow $Ca^{2+}$ chelator, EGTA-AM, was used. In the current study, extracellular $Ca^{2+}$ was removed and EGTA was included in the bath solution. Moreover, intracellular stores of $Ca^{2+}$ were depleted using CPA and the fast $Ca^{2+}$ chelator, BAPTA-AM. The finding that WT neurons, in $Ca^{2+}$-free conditions, display a similar RRP recovery rate as compared to KO neurons in normal ($Ca^{2+}$) conditions, along with the inability of the syt 7 4D/N mutant to rescue recovery of the RRP in the presence of $Ca^{2+}$ (*Figure 6*), strongly indicate that syt 7 is a $Ca^{2+}$-sensor in the $Ca^{2+}$-dependent SV replenishment pathway.

CaM is a ubiquitous $Ca^{2+}$-binding protein with multiple functions in the SV cycle, including SV replenishment (*Sakaba and Neher, 2001*; *Hosoi et al., 2007*). Remarkably, when applied to WT hippocampal neurons, a CaM antagonist precisely phenocopies the defects in SV replenishment apparent in KO neurons. Moreover, we discovered an avid and direct $Ca^{2+}$-promoted interaction between CaM

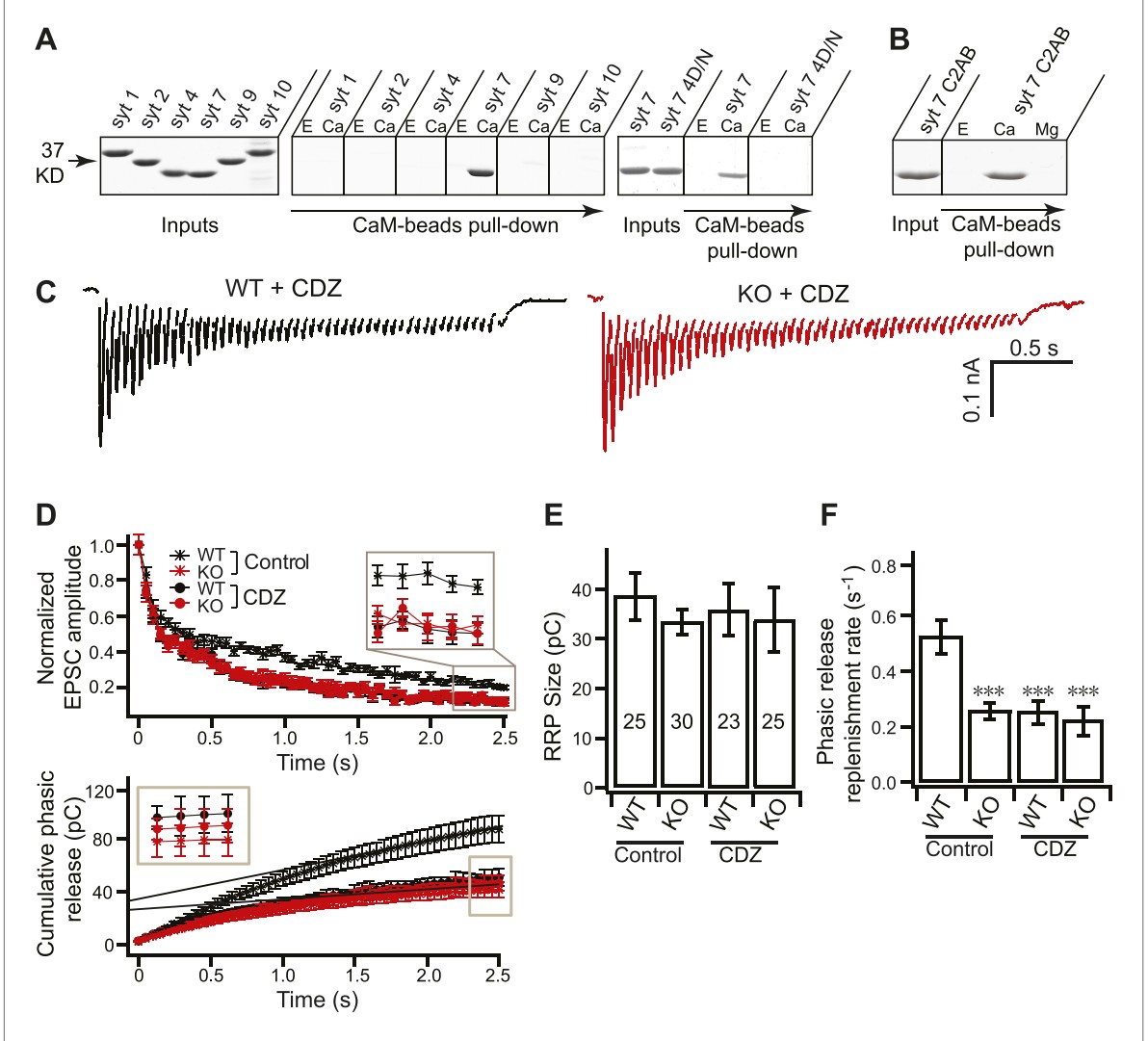

**Figure 7**. Syt 7 cooperates with CaM to regulate $Ca^{2+}$-dependent SV replenishment. (**A**) Immobilized CaM efficiently pulled-down the C2AB domain of syt 7, but not syt 1, 2, 4, 9 or 10, in the presence, but not the absence, of $Ca^{2+}$; the syt 7 4D/N mutant failed to bind. E, 2 mM EGTA; Ca, 1 mM free $Ca^{2+}$. Spaces between gels indicate samples that were run separately. Input corresponds to 25% of the starting material; 50% of the bound material was loaded per lane; no binding was observed when CaM was omitted from the beads (data not shown). (**B**) $Mg^{2+}$ fails to promote binding of syt 7 C2AB to CaM. E, 2 mM EGTA; Ca, 1 mM free $Ca^{2+}$; Mg, 1 mM free $Mg^{2+}$. Gels were loaded as in panel **A**. (**C**) Representative EPSCs traces (20 Hz/2.5 s) recorded from WT (black) and KO (red) neurons in the presence of CDZ. (**D**) Plot of cumulative total phasic release charge vs time in the presence of CDZ. Data were analyzed as described in *Figure 3C*. (**E**) The size of the RRP was the same among all groups (in control: WT: 38.70 ± 4.78 pC; KO: 33.49 ± 2.48 pC; in CDZ treated neurons: WT: 36.85 ± 5.35 pC; KO: 34.77 ± 6.74 pC). (**F**) CDZ treatment reduced SV replenishment in WT neurons (0.24 ± 0.04 $s^{-1}$ vs 0.52 ± 0.06 $s^{-1}$ for controls), but had no effect on KO neurons (0.21 ± 0.05 $s^{-1}$ vs 0.26 ± 0.03 $s^{-1}$ for controls). The n values indicate the number of neurons, obtained from three independent litters of mice, used for these experiments. All data shown represent mean ± SEM. ***p<0.001, analyzed by one-way ANOVA, compared to WT.

The following figure supplements are available for figure 7:

**Figure supplement 1**. CDZ inhibits SV replenishment during tonic transmission in WT neurons.

and syt 7; in contrast, syt 1, 2, 4, 9 and 10 failed to bind to CaM in response to $Ca^{2+}$, and loss of these latter isoforms has not been associated with enhanced synaptic depression (*Maximov and Sudhof, 2005*; *Pang et al., 2006*; *Sun et al., 2007*; *Xu et al., 2007*; *Dean et al., 2009*; *Liu et al., 2009*; *Cao et al., 2011*; *Dean et al., 2012a*). These findings underscore both the specificity of the syt 7•CaM interaction, as well as the specific role played by syt 7 during synaptic depression. Thus, we propose that the $Ca^{2+}$-sensor for replenishment consists of a complex formed between two conserved $Ca^{2+}$-binding proteins, syt 7 and CaM.

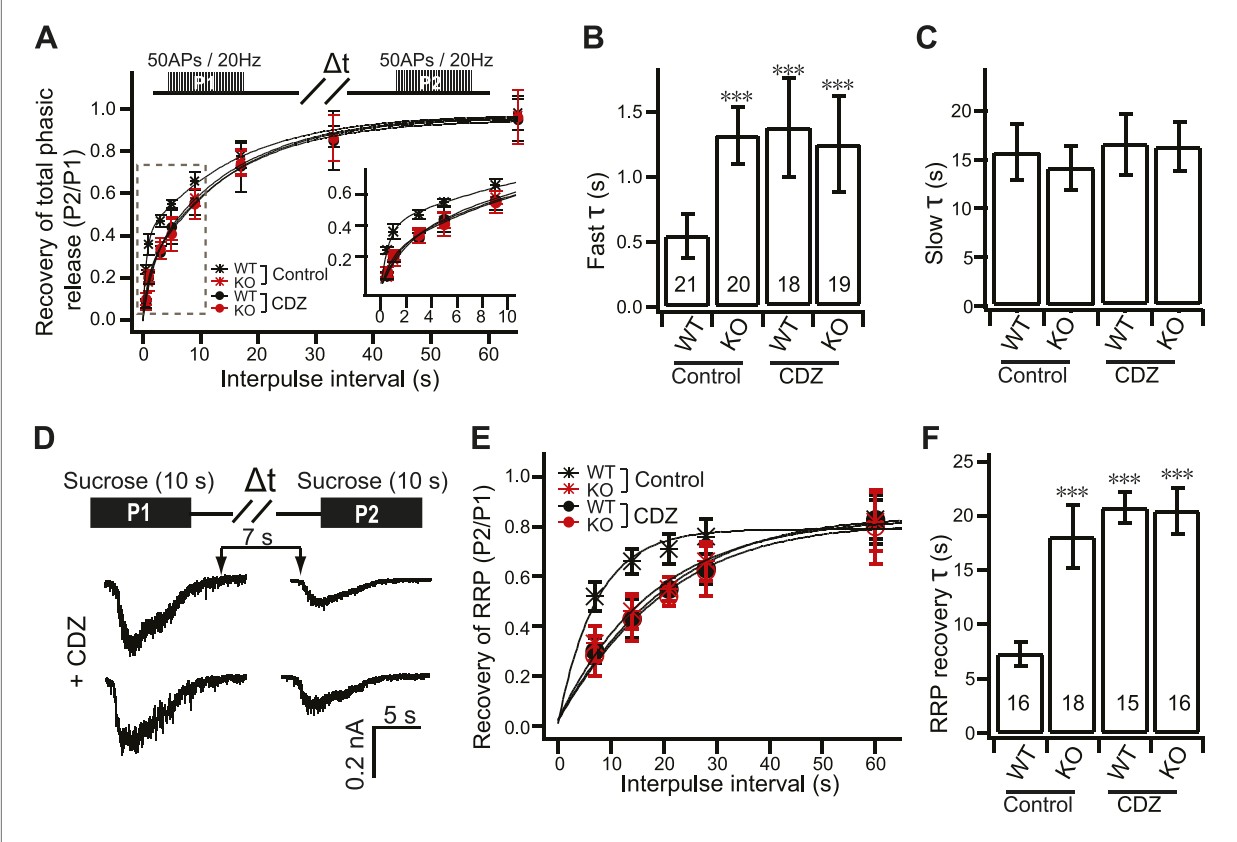

**Figure 8**. CDZ inhibits RRP recovery in WT neurons, phenocopying the KO. (**A**) Time course for the recovery of phasic release upon HFS; data were collected and analyzed as described in *Figure 4A*. The inset panel shows the expanded early phase of the plot. (**B**) CDZ slowed the time course of fast recovery of the RRP in WT neurons (1.35 ± 0.37 s vs 0.55 ± 0.17 s for control), phenocopying the KO neurons, which were not affected by CDZ (1.22 ± 0.36 s vs 1.32 ± 0.22 s for control). (**C**) The time course of slow recovery of the RRP was the same between WT and KO neurons with or without CDZ (WT: 16.02 ± 3.02 s vs 15.72 ± 2.85 s for controls; KO: 15.78 ± 2.5 s vs 14.13 ± 2.24 s for controls). (**D**) Typical responses to sequential (7 s) applications of sucrose to WT and KO neurons in 2 mM $[Ca^{2+}]_o$ in the presence of CDZ. The upper diagram illustrates the protocol used ($\Delta t$ = 7, 14, 21, 28 and 60 s). (**E**) Time course for the recovery of the RRP after depletion with hypertonic sucrose in 2 mM $[Ca^{2+}]_o$. Data were fitted with a single exponential function to calculate the recovery time constant (F). (**F**) CDZ increased the time constant for recovery of the RRP in WT neurons (20.69 ± 1.43 s vs 7.29 ± 1.05 s for controls), but had no effect on KO neurons (20.37 ± 2.1 s vs 18.03 ± 2.87 s for controls). The n values indicate the number of neurons, obtained from three independent litters of mice, used for these experiments. All data shown represent mean ±SEM. ***$p<0.001$, analyzed by one-way ANOVA, compared to WT.

The following figure supplements are available for figure 8:

**Figure supplement 1**. CDZ impairs recovery of tonic transmission in WT neurons.

**Figure supplement 2**. CDZ does not compete with syt 7 for binding to CaM.

From the data reported here, neuronal functions for the only three syt isoforms found in all metazoans (isoforms 1, 4 and 7; *Barber et al., 2009*) have been determined. In presynaptic terminals, syt 1 acts to directly trigger rapid SV exocytosis (*Geppert et al., 1994*; *Koh and Bellen, 2003*; *Nishiki and Augustine, 2004*; *Liu et al., 2009*), syt 4 regulates the release of brain derived neurotrophic factor to modulate synaptic function (*Dean et al., 2009, 2012b*), and syt 7 functions as a $Ca^{2+}$ sensor for SV replenishment during and after HFS.

In a previous study it was concluded that neurotransmission was completely normal in KO inhibitory neurons, and that syt 7 did not regulate asynchronous synaptic transmission (*Maximov et al., 2008*). In that study, there was a slight reduction in synaptic transmission in KO neurons during high, but not low, frequency stimulation, suggestive of a defect in replenishment. This effect was less pronounced as compared to our observations; the reasons for this difference are unknown, but might involve differences between the inhibitory and excitatory neurons used in these studies. Moreover, a more

recent report from the same group (*Bacaj et al., 2013*), using a KD approach, reached the contradictory conclusion that syt 7 plays a role in asynchronous evoked release. However, loss of syt 7, by either KD or KO, had no effect on asynchronous release (*Maximov et al., 2008*; *Bacaj et al., 2013*); the only case where a loss of release was observed was when syt 7 was knocked down (and not knocked out) in a syt 1 KO background. So, the physiological relevance of this observation is unclear. Moreover, in *Bacaj et al. (2013)*, there was, again, a reduction in release only at late, but not early stages, in the stimulus train in KO neurons. We argue that this result should be interpreted as defective SV replenishment, because loss of asynchronous release should occur for all evoked responses, early and late. Furthermore, impaired replenishment of large dense core vesicles (LDCV) was also reported in KO chromaffin cells (*Schonn et al., 2008*), consistent with our current study. Thus, syt 7 might serve as a ubiquitous $Ca^{2+}$ sensor for replenishment of vesicles, including LDCVs and SVs.

At present, it remains unknown as to how syt 7 and CaM regulate SV replenishment. We observed that c-myc tagged WT syt 7, as well as the 4D/N mutant, were enriched in synapses, but were not co-localized with SV markers (*Figure 3—figure supplement 2*). These findings raise the possibility that syt 7 may be targeted to, and thereby act at the plasma membrane, as proposed by *Sugita et al. (2001)*, to drive replenishment by either conveying SVs from reserve pools to the RRP (*Kuromi and Kidokoro, 1998*; *Murthy and Stevens, 1999*; *Ikeda and Bekkers, 2009*), or clearing vesicle release sites (*Kawasaki et al., 2000*; *Wu et al., 2009*; *Neher, 2010*). Using a pHluorin tag, we found that syt 1 and 7 have similar cell surface fractions; while significant levels of each protein are present in the plasma membrane, the largest pools of syt 7 and syt 1 are on internal organelles that acidify (*Figure 3—figure supplement 3*), consistent with earlier work localizing syt 7 to LDCVs and lysosomes in neuro-endocrine cells and cells in the immune system (*Fukuda et al., 2004*; *Wang et al., 2005*; *Czibener et al., 2006*). Clearly, syt 7 is not selectively targeted to the plasma membrane (though, again, like all syt isoforms, some fraction is present in the plasma membrane [*Dean et al., 2012a*]). Moreover, pHluorin-syt 7 does not recycle with SV-like kinetics, even after repetitive stimulation (*Dean et al., 2012a*), excluding the possibility that syt 7 traffics to SVs before, during, or after stimulation. So, syt 7 is likely to regulate replenishment either from the plasma membrane, or from LDCVs or lysosomes.

In molecular terms, the emerging view is that CaM regulates SV replenishment via interactions with both Munc13-1 (*Lipstein et al., 2013*) and syt 7. Future studies are needed to determine whether CaM can physically interact with both proteins simultaneously, or whether CaM forms mutually exclusive complexes in the replenishment pathway.

## Materials and methods

### Cell culture

KO mice were provided by NW Andrews (College Park, MD). Primary hippocampal cultures were prepared as described previously (*Liu et al., 2009*). Briefly, heterozygous KO mice were bred, and hippocampal neurons from newborn pups (postnatal day 0) were isolated. Neurons were plated at low density (~5000 cells/cm²) on poly-lysine coated coverslips (Carolina Biologicals, Burlington, NC), and cultured in Neurobasal media supplemented with 2% B-27 and 2 mM Glutamax (Life Technologies, Great Island, NY). Tails from KO pups were kept for genotyping, and electrophysiological recordings from KO and WT littermate neurons were compared.

All procedures involving animals were performed in accordance with the guidelines of the National Institutes of Health; the protocol used (M01221-0-06-11) was approved by the Institutional Animal Care and Use Committee of the University of Wisconsin–Madison.

### Molecular biology

cDNA encoding rat syt 1 and syt 4 were provided by TC Südhof (Stanford, CA) and H. Herschman (Los Angeles, CA), respectively. cDNA encoding mouse syt 2, 7, 9, and 10 were provided by M Fukuda (Saitama, Japan). We note that the D374 mutation in the original syt 1 cDNA was corrected by replacement with a glycine residue (*Desai et al., 2000*). cDNA encoding sypHy was provided by E Kavalali (Dallas, TX).

The syt 7 4D/N mutant harbors a D225,227,357,359N quadruple mutation in which four acidic $Ca^{2+}$-ligands (two in each C2 domain) were neutralized, thereby disrupting $Ca^{2+}$-binding activity.

Because specific, high affinity syt 7 antibodies are not currently available, a c-myc tag, TGGAGCAGAAGCTGATCAGCGAGGAGGACCTGAACGGAATT, was added to the N-terminus of

WT and syt 7 4D/N mutant to detect and localize the exogenously expressed proteins. C-myc tagged syt 1 was used as a control. A signal sequence (ss): ATGGACAGCAAAGGTTCGTCGCAGAAAG GGTCCCGCCTGCTCCTGCTGCTGGTGGTGTCAAATCTACTCTTGTGCCAGGGTGTGGTCTCCGA, was appended onto the N-terminus of the c-myc tag to ensure translocation of the fusion proteins (ss-myc-syt 1, ss-myc-syt 7 and ss-myc-syt 7 4D/N) into the endoplasmic reticulum (*Dean et al., 2012a*). These constructs were subcloned into the pLox Syn-DsRed-Syn-GFP lentivirus vector (provided by F Gomez-School [Spain]) using BamH I and Not I, thereby replacing the DsRed sequence. The GFP was used to identify infected neurons. For pHluorin imaging, sypHy (*Kwon and Chapman, 2011*) was sub-cloned into this vector, again using BamH I and Not I to replace DsRed. GFP in the pLox vector was replaced by mCherry to avoid spectral overlap with pHluorin.

## Infection, transfection and immunostaining of hippocampal neurons

For electrophysiological recordings, WT syt 7 or the 4D/N mutant were expressed in KO neurons at 5 days in vitro (DIV) using lentivirus as described previously (*Dong et al., 2006*).

For the pHluorin imaging experiments, neurons were grown on 12 mm coverslips in 24-well plates and transfected using the calcium phosphate method at 3–4 days in vitro (DIV) as described previously (*Dresbach et al., 2003*; *Dean et al., 2012a*). Prior to transfection, medium was removed, saved, and replaced with 500 µl Optimem (Life Technologies, Great Island, NY) and incubated for 30–60 min. 105 µl of transfection buffer (274 mM NaCl, 10 mM KCl, 1.4 mM $Na_2HPO_4$, 15 mM glucose, 42 mM HEPES, pH 7.06) was added drop-wise to a 105 µl solution containing 7 µg of DNA and 250 mM $CaCl_2$, with gentle vortexing. This mixture was incubated for 20 min at room temperature (RT); 30 µl was added per well and neurons were further incubated for 60–90 min. Medium was removed, cells were washed three times in Neurobasal medium, and the saved medium was added back to the transfected cells. Neurons were used for imaging between 12–17 DIV.

For the immunostaining experiments, neuronal cultures were immunostained with a mouse mono-clonal antibody directed against c-myc (EMD Millipore, Bilerica, MA) to determine the localization of exogenously expressed c-myc-tagged syt 1, syt 7 and the syt 7 4D/N mutant. A polyclonal rabbit anti-synapsin antibody (Synaptic System, Germany) was used to localize synaptic vesicles in presynaptic terminals. 12–17 DIV neurons were fixed with 4% paraformaldehyde in PBS, permeabilized with 0.15% Triton X-100 for 30 min, blocked in 10% goat serum plus 0.1% Triton X-100, stained with primary anti-bodies for 2 hr, washed with PBS three times, and then incubated with either Cy3-tagged anti-mouse or Alexa 647-tagged anti-rabbit (Jackson ImmunoResearch Laboratories, West Grove, PA) secondary antibodies for 1 hr at room temperature. Coverslips were then mounted in Fluoromount (Southern Biotechnology, Birmingham, AL) and images of were acquired on an Olympus FV1000 upright confocal microscope (Japan) with a 60× 1.10 numerical aperture water-immersion lens. To quantify the degree colocalization, images were imported into ImageJ (NIH) for analysis. The Pearson's coefficient for colo-calization was calculated using the intensity correlation analysis plugin for ImageJ.

## Electrophysiology

As described previously (*Liu et al., 2009*), whole-cell voltage-clamp recordings of mEPSCs, evoked EPSCs, and hypertonic sucrose responses were carried out at RT using an EPC-10/2 amplifier (HEKA, Germany) from neurons at 12-17 DIV. The pipette solution consisted of, in mM: 130 K-gluconate, 1 EGTA, 5 Na-phosphocreatine, 2 Mg-ATP, 0.3 Na-GTP, and 10 HEPES, pH 7.3 (290 mOsm). The extracellular solution consisted of (mM): 140 NaCl, 5 KCl, 1 $MgCl_2$, 10 glucose, 10 HEPES, pH 7.3 (300 mOsm), 0.05 D-2-amino-5-phosphonopentanoate (D-AP5), 0.1 picrotoxin. For mEPSC recordings, 2 mM $Ca^{2+}$ was included, and 1 µM tetrodotoxin (TTX) was added to the extracellular solution. For evoked EPSCs, 5 mM QX-314 (lidocaine N-ethyl bromide) was added to the pipette solution. Various concentrations of $[Ca^{2+}]_o$ (in mM: 0.5, 2, 5 and 10) were used for recording single AP evoked EPSCs. Ten mM $[Ca^{2+}]_o$ was used for the HFS recordings in order to maximize the first response. To evoke EPSCs, a bipolar electrode was placed against the soma, and used to trigger action potentials via a 1 ms minus 20 V pulse, as described previously (*Liu et al., 2009*). Release was recorded as a postsynaptic response via a whole-cell patch electrode in a connected neuron.

Neurons were voltage clamped at −70 mV. Only cells with series resistances of <15 M, with 70–80% of this resistance compensated, were analyzed. Currents were acquired using PATCHMASTER software (HEKA, Germany), filtered at 2.9 kHz, and digitized at 10 kHz. Data were analyzed using MiniAnalysis software (Synaptosoft, Decatur, GA), Clampfit (Molecular Devices, Sunnyvale, CA), and Igor (Wavemetrics, Portland, OR).

For hypertonic sucrose experiments, we puffed sucrose over the entire area viewed under a 40× objective lens, which includes virtually all presynaptic boutons contacting the patched cells. 500 mM sucrose was applied to neurons, using an air pressure system (PicoSpritzer III, Parker Hannifin, Cleveland, OH), for 10 s to empty the RRP of vesicles. After a quick wash (Masterflex C/L, Cole–Parmer, Vernon Hills, IL), a second puff of sucrose was applied for another 10 s at various interpluse intervals. The recovery ratio was calculated by dividing the total charge elicited by the second puff of sucrose by the total charge elicited by the first puff. For the sucrose experiments in the presence of physiological $[Ca^{2+}]_o$, the extracellular solution included 2 mM $Ca^{2+}$. For the $Ca^{2+}$-free condition, $[Ca^{2+}]_o$ was replaced with 5 mM EGTA, and 25 µM BAPTA-AM (30 mM stock in DMSO) and 30 µM CPA (100 mM stock in DMSO) were added to the bath solution; cultures were incubated for 30 min before recording to allow BAPTA-AM and CPA to exert their effects.

For the recordings using bafilomycin, 1 µM bafilomycin (1.6 mM stock in DMSO) was added to the bath solution, and incubated for 3 min prior to recording. Neurons that were treated with bafilomycin were used within 20 min. For the CDZ experiments, 20 µM CDZ (100 mM stock in DMSO) was added to the bath solution and neurons were incubated for 30 min prior to recording. In the CTZ and KYN experiments, both 50 µM CTZ (100 mM stock in DMSO) and 100 µM KYN (75 mM stock in DMSO) were added to the bath.

All chemicals were purchased from Sigma–Aldrich.

## PHluorin imaging experiments

PHluorin imaging was performed as described previously (*Kwon and Chapman, 2011*; *Dean et al., 2012a*). Neurons were transferred to a live cell imaging chamber (Warner Instruments, Hamden, CT) and continuously perfused with bath solution (140 mM NaCl, 5 mM KCl, 2 mM $CaCl_2$, 2 mM $MgCl_2$, 10 mM HEPES, 10 mM glucose, 0.1 mM picrotoxin, 50 µM D-AP5, 10 µM CNQX; 310 mOsm, pH 7.4) at RT. The transfected cells were identified via mCherry fluorescence. For field-stimulation, 1 ms voltage pulses (70 V) were delivered using an SD9 stimulator (Natus Neurology, Warwick, RI) connected to two parallel platinum wires spaced by 10 mm in the imaging chamber (Warner Instruments, Hamden, CT). Time-lapse images were acquired at 1 s intervals, with 484ex/517em, on an inverted microscope (Nikon TE300, Japan) with a 1.42 NA 100× immersion oil objective under illumination with a Xenon light source (Lambda DG4, Sutter Instrument, Novato, CA). Fluorescence changes at individual boutons were detected using a Cascade 512II EMCCD camera (Roper Scientific, Trenton, NJ); data were collected and analyzed offline using MetaMorph 6.0 software. A baseline of 10 images was collected before applying high-frequency field stimulation. A typical exposure time was 400 ms for pHluorin. Puncta that did not exhibit any lateral movement during image acquisition were chosen for analysis.

For pHluorin-syt 7 and pHluorin-syt 1 quenching/de-quenching experiments, fast solution exchanges were achieved using an MBS-2 perfusion system (MPS-2, World Precision Instruments, Sarasota, FL). The acidic solution, with a pH of 5.5, was prepared by replacing HEPES, in the bath saline used for perfusion, with 2-[*N*-morpholinoethane sulphonic acid (pKa = 6.1). An ammonium chloride solution (pH 7.4) was prepared by substituting 50 mM NaCl in bath saline with $NH_4Cl$, while all other components remained unchanged.

## Recombinant proteins

Recombinant cytoplasmic domains of syt isoforms were generated as either GST fusion proteins, or his6-tagged proteins. For GST fusion proteins, cDNA encoding the tandem C2 domains of syt 1 (residues 96–421), WT (residues 134–403) or the 4D/N mutant form of syt 7, and cDNA encoding the isolated C2A (residues 134–263 ) or C2B (residues 243–403) domain of WT (syt 7 C2A; syt 7 C2B) or the $Ca^{2+}$-ligand mutant forms of each C2-domain (syt 7 C2A 2D/N, D225,227N; syt 7 C2B 2D/N, D357,359N) of syt 7, were expressed as glutathione S-transferase (GST) fusion proteins and purified using glutathione-Sepharose beads (GE healthcare), as described previously (*Bhalla et al., 2008*). The GST tag was removed via thrombin cleavage. For his6-tagged proteins, cDNA encoding the tandem C2 domains of syt 1 (residues 96–421), syt 2 (residues 139–423), syt 4 (residues 152–425), syt 7 (residues 134-403), syt 9 (residues 104–386), and syt 10 (residues 223–501) were subcloned into either pTrc-HisA (Invitrogen) or pET28a (Novagen) generating an N-terminal $His_6$-tag. $His_6$-tagged proteins were purified using Ni Sepharose (GE healthcare), as described previously (*Bhalla et al., 2008*)

## CaM pull-down assay

Aliquots of CaM agarose (EMD Millipore, Bilerica, MA), containing 20 µg of CaM, were mixed with 2 µM of the indicated syt fragment in 150 µl of Dulbecco's Phosphate Buffered Saline (DPBS) plus

0.5% Triton X-100 in the presence of 2 mM EGTA or 1 mM free $Ca^{2+}$ or 1 mM free $Mg^{2+}$. The mixture was incubated with rotation at 4°C for 1 hr, after which the CaM beads were collected via centrifugation at 2000×$g$ for 50 s, and washed three times with DPBS. Beads were boiled in sample buffer for 5 min, and then subjected to SDS-PAGE. Proteins were visualized by staining with Coomassie blue. In *Figure 8—figure supplement 2*, samples were incubated for 2 hr at 37°C.

## Circular Dichroism (CD) experiments

CD spectra were obtained using a Model 420 Circular Dichroism Spectrophotometer (Aviv Biomedical) at 20°C using a 1 mm path length cuvette in 50 mM $Na_2HPO_4$ (pH 7.4); the protein concentration was 0.4 mg/ml.

## Isothermal Titration Calorimetry (ITC) experiments

ITC was carried-out using a MicroCal iTC$_{200}$ calorimeter (GE Healthcare, Madison, WI). WT and the $Ca^{2+}$-ligand (2D/N) mutant form of each isolated C2-domain of syt 7 were dialyzed overnight against ITC buffer (50 mM HEPES, 200 mM NaCl, 10% glycerol, pH 7.4) in the presence of Chelex-100 resin. Heat of binding was measured from 11 serial injections of $Ca^{2+}$ (3.5 μl of 7 mM $Ca^{2+}$ for C2A and 15 mM $Ca^{2+}$ for C2B; higher [$Ca^{2+}$] was needed to saturate C2B) into a sample cell containing 1.5 mg/ml (100 μM) protein at 25°C. Data were corrected for heat of dilution.

## Statistical analysis

Electrophysiology and imagining data were obtained from three independent litters of mice, unless otherwise indicated. Data were pooled for statistical analysis, as described in previous studies (*Xu et al., 2007*; *Liu et al., 2009*; *Kwon and Chapman, 2011*). All data are presented as the mean ± SEM and significance was determined using the Student's $t$ test, one-way ANOVA, or two-way ANOVA (*$p < 0.05$, **$p < 0.01$, and ***$p < 0.001$), as appropriate (Prism 6, Graphpad Software).

## Acknowledgements

We thank members of the Chapman laboratory for helpful discussions, L Roper for providing WT and KO neurons, and M Dong for providing the myc-tagged versions of syt 1 and 7. ERC is an Investigator of the Howard Hughes Medical Institute.

## Additional information

### Funding

| Funder | Grant reference number | Author |
|---|---|---|
| National Institutes of Health | MH 61876 | Huisheng Liu, Hua Bai, Enfu Hui, Lu Yang, Chantell S Evans, Zhao Wang, Sung E Kwon, Edwin R Chapman |
| Howard Hughes Medical Institute | | Enfu Hui, Edwin R Chapman |

The funder had no role in study design, data collection and interpretation, or the decision to submit the work for publication.

### Author contributions

HL, Designed and performed experiments, Analyzed data, Co-wrote the paper; HB, EH, ZW, Performed and analyzed the biochemical experiments; LY, Performed and analyzed the imaging experiments; CSE, Performed the CD and ITC experiments; SEK, Performed the pHluorin-syt quenching experiment; ERC, Designed experiments, Co-wrote the paper

### Ethics

Animal experimentation: All procedures involving animals were performed in accordance with the guidelines of the National Institutes of Health; the protocol used (M01221-0-06-11) was approved by the Institutional Animal Care and Use Committee (IACUC) of the University of Wisconsin–Madison.

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
