## [Decision Letter]

Thank you for sending your work entitled “Synaptotagmin VII Functions as a Ca^2+^-sensor for Synaptic Vesicle Replenishment” for consideration at *eLife*. Your article has been favorably evaluated by a Senior editor and 3 reviewers, one of whom is a member of our Board of Reviewing Editors.

The Reviewing editor and the other reviewers discussed their comments before we reached this decision, and the Reviewing editor has assembled the following comments to help you prepare a revised submission.

This is a well-done piece of work indicating that synaptotagmin VII has a specific role in synaptic vesicle mobilization. The analyses rely almost completely on a physiological characterization of synaptotagmin VII knockout mice, and data interpretation is greatly aided by the selective effect with no observable change in the response to a single action potential or pool size. In addition, the results are consistent with previous work showing little effect of SytVII on synaptic transmission. An important element of this study is the experiments to rescue the defect with either wild type SytVII or with a mutant version of SytVII that lacks calcium binding (4D/N). An analysis of recovery from depression supports the conclusion that SytVII is required for the fast phase of vesicle recovery, a deficit of ∼50% is resolved. In general, the experiments are nicely performed and clearly documented.

Despite the quality of the data, all three reviewers raised similar concerns with the manuscript that must be addressed in order for this work to be considered further at *eLife*. These concerns must be addressed.

1) The authors have in no way addressed the fact that in a 2008 publication from the Sudhof lab (Maximov et al., PNAS) the claim is that there is in fact no real depression in synaptic depression during HFS. Interestingly the authors actually agree on many points, i.e., no change in RRP, no change in PPR etc. However as this is really the crux of this paper and the two studies on the face of it simply do not agree I think it is imperative that this be explained, or examined. There is a possible trivial explanation, which is that the Sudhof report was on inhibitory neurons. If that is the explanation then is lessens the generality of this finding. This must be addressed in the text or through further experimentation.

2) It is very difficult to understand how a protein that does not localize to either the plasma membrane or synaptic vesicles influences the replenishment of synaptic vesicles. The authors suggest that the known role of syt VII in dense core vesicle release may contribute, presumably by releasing a factor that influences synaptic vesicle recycling. However, the loss of syt VII does not eliminate regulated exocytosis of chromaffin granules, and it is harder to understand how a partial defect would produce this effect on synaptic transmission, particularly since the defect occurs as early as the second action potential in a train. This issue lies at the very heart of this study. As such, this issue has to be introduced in the Introduction and should be clearly and openly discussed. In addition, the authors should attempt additional experimentation to confirm that SytVII does indeed reside on an internal membrane and provide further experimentation to address how SytVII could specifically influence recovery. The interation with calmodulin is interesting, but it remains a protein interaction without a clearly understood relevance to vesicle recovery.

3) The work provides no mention of the number of animals or samples used for any of the experiments, in either the Methods or the figure legends. The legend to Figure 1 mentions three independent trials, but it is unclear what that means (animals, slices, cells), or its relationship to the statistics. This is a serious deficiency and a low n could greatly reduce confidence in this entire study. Similarly, the imaging in Figure 5 presents unnormalized changes in fluorescence, and it is very hard to evaluate this either-it would be particularly difficult to combine data like this from multiple independent experiments, again raising concern about n.

4) The authors demonstrate that the D/N mutations in the C2 domain do not rescue, consistent with the need for this tagmin to bind calcium for its function. However the authors never actually show that this protein is actually present at synapses when re-expressed as a mutant. This should be addressed experimentally.

---

## [Author Response]

*1) The authors have in no way addressed the fact that in a 2008 publication from the Sudhof lab (Maximov et al., PNAS) the claim is that there is in fact no real depression in synaptic depression during HFS. Interestingly the authors actually agree on many points, i.e. no change in RRP, no change in PPR etc. However as this is really the crux of this paper and the two studies on the face of it simply do not agree I think it is imperative that this be explained, or examined. There is a possible trivial explanation, which is that the Sudhof report was on inhibitory neurons. If that is the explanation then is lessens the generality of this finding. This must be addressed in the text or through further experimentation*.

We thank the referees for raising this issue. There are numerous self-contradictory reports from the Sudhof lab concerning the synaptotagmins (syt), and in most cases we are unable to explain these contradictions. Specifically, regarding syt 7: a) the Sudhof lab reported that syt 1 and 7 had the same Ca^2+^- affinity for Ca^2+^-dependent binding to membranes (Li et al., 1995), and reported that syt 7 has a much higher Ca^2+^-affinity for membrane binding, as compared to syt 1 (Sugita et al., 2002).

b) In 2001, the Sudhof lab reported that: “*We now propose that synaptotagmin VII functions as a plasma membrane Ca*^*2+*^
*sensor in synaptic exocytosis complementary to vesicular synaptotagmins*” (Sugita et al., 2001), but later reported that: “…*Ca*^*2+*^
*binding by synaptotagmin-VII likely does not regulate SV exocytosis…*” (26). [We also note that the Sudhof lab reported that syt 1 must be glycosylated in order to be targeted to SVs, and the reason why syt 7 was targeted to the plasma membrane was because it was not glycosylated (Han et al., 2004). However, we found that glycosylation plays no role in the targeting of syt 1 to SVs, and that syt 7 is not selectively targeted to the plasma membrane (Dean et al., 2012; Kwon and Chapman, 2012).]

c) During review of our syt 7 manuscript, the Sudhof lab published a new syt 7 paper (1) where they now state: “*We identify a selective essential function for Syt 7 in asynchronous release.*” This contradicts their earlier paper (26), in which they state, again, that “*…Ca*^*2+*^
*binding by synaptotagmin-VII likely does not regulate SV exocytosis…*”, and that syt 7 plays no role in asynchronous release.

In [1], the authors used a knock-down (KD) approach to conclude that syt 7 plays a role in asynchronous release. The authors propose that compensatory mechanisms, that result from the knock-out (KO) of syt 7, preclude functional effects (and hence explain the lack of an effect in [26], where syt 7 KOs were used), and that a KD approach should be used to study syt 7. This claim predicts that KD of syt 7 should decrease asynchronous release in otherwise WT neurons, but this was not the case: the authors saw no effect on asynchonrous release when syt 7 was knocked-down in otherwise WT neurons (1). So, there is complete agreement that KO or KD of syt 7 has no effect on asynchronous release from mouse central neurons.

In summary, the physiological relevance of the new proposal from the Sudhof lab, that syt 7 regulates asynchronous release, is unclear, as it occurred only when syt 7 was knocked-down, but not knocked-out, in a syt 1 KO background. Moreover, in the syt 7 KD/syt1 KO study, a key control – hypertonic sucrose (to determine the size and existence of releasable vesicle pools) – was missing, so the reported phenotype could simply be due to a loss of releasable SVs, and the published data set cannot be interpreted.

Getting back to the specific point raised by the referees, we note that in Figure 5B of [26], there was in fact a slight reduction in synaptic transmission in syt 7 KO neurons during high, but not low frequency stimulation, suggestive of a defect in vesicle replenishment. These results are consistent with our findings, although the magnitude of the effect was smaller in Maximov et al (2008). Moreover, in [1], there is a reduction in release at later, but not early, stages in the stimulus train, which we argue should be interpreted as defective SV replenishment, not the “suppression” of “asynchronous” release posited by the authors (1). Loss of asynchronous release should occur for all evoked responses, early and late, but in Figure 7c of [1] this is not the case: a loss of charge transfer became apparent only at late stages in the stimulus train in syt 7 KO neurons, again consistent with our model that posits that syt 7 plays in role in SV replenishment, but not in asynchronous release.

Furthermore, impaired replenishment of large dense core vesicles (LDCVs) was also reported in syt 7 KO chromaffin cells, where the authors state:

“*Replenishment was inhibited in KO cells*” (39). Therefore, we hypothesize that syt 7 is a ubiquitous Ca^2+^ sensor for vesicle replenishment, contributing to the replenishment of both LDCVs and SVs.

We have included a short discussion of the [26] paper and the [1] paper in “Discussion” section of the revised manuscript, to help to clarify these points.

*2) It is very difficult to understand how a protein that does not localize to either the plasma membrane or synaptic vesicles influences the replenishment of synaptic vesicles*.

Our new pHluorin-syt 7 and -syt 1 quenching/de-quenching experiments confirmed (please see also (Dean et al., 2012)) that the surface and internal fractions of these isoforms were similar (revised Figure 3—figure supplement 3), thus syt 7 is not selectively targeted to the plasma membrane (as discussed above). Clearly, some fraction of each proteins is present in the plasma membrane; in general, all syt isoforms are present, to some degree, in the plasma membrane (Dean et al., 2012), and could, in principle, regulate release from this position.

How any of the proteins, shown to regulate replenishment (CaM, munc13-1, syt 7), operate in this process remains to be determined. The first goal is to identify all the factors, and then to figure out how they work together to regulate replenishment, thereby elucidating the underlying molecular mechanisms. At present, few factors have been identified. Importantly, our study provides an entirely new, novel, and critical regulatory element in the replenishment pathway, syt 7. Rather than engage in extensive speculation as to how syt 7 regulates replenishment, we now state, in the “Discussion” section of the revised manuscript, that:

“… So, syt 7 is likely to regulate replenishment either from the plasma membrane, or from LDCVs or lysosomes. In molecular terms, the emerging view is that CaM regulates SV replenishment via interactions with both munc13-1 (23) and syt 7. Future studies are needed to determine whether CaM can physically interact with both proteins simultaneously, or whether CaM forms mutually exclusive complexes in the replenishment pathway.”

We should also note that “replenishment” is an operational definition, and the physical basis that underlies this process is currently the subject of debate: one model posits that replenishment is mobilization from a reserve or reluctant pool to a recycling pool (18; 21; 28), while another model posits that replenishment is limited by clearance of release sites (19; 29; 47). So, as we identify key components of the replenishment pathway, the hope is to determine whether these components act to regulate mobilization of SVs or clearance of release sites, and these experiments are likely to require new innovations to address issues such as the movement of SV components away from active zones etc.

*The authors suggest that the known role of syt VII in dense core vesicle release may contribute, presumably by releasing a factor that influences synaptic vesicle recycling. However, the loss of syt VII does not eliminate regulated exocytosis of chromaffin granules, and it is harder to understand how a partial defect would produce this effect on synaptic transmission, particularly since the defect occurs as early as the second action potential in a train. This issue lies at the very heart of this study. As such, this issue has to be introduced in the Introduction and should be clearly and openly discussed*.

If the referees are referring to the electrophysiological recordings under HFS, there seems to be a misunderstanding as the second AP gave rise to normal release (please see the PPR in Figure 2). If the referees are referring to the experiment shown in Figure 5, we note that the second data point in panel C corresponds to 1 second of stimulation at 20 Hz, not the 2^nd^ stimulation. The normal release observed at early stages, in conjunction with the deficiencies observed at later stages in syt 7 KO neurons, are indicative of reductions in SV replenishment.

We understand the point raised by the referees: ∼1 second provides little time for LDCV release to modify synaptic transmission, but it is also possible that previous release of the putative syt 7 LDCV is permissive for SV replenishment. But this example begins to illustrate the problem with speculation: since so little is known concerning replenishment it seems, to us, that too much speculation is not appropriate. Rather than speculate, we simply state that syt 7 must execute its function from where it is located: at the plasma membrane, or on internal vesicles that likely correspond to LDCVs and lysosomes. So, we have simplified the text to some extent to minimize speculation, as detailed in our response to point #2 above.

*In addition, the authors should attempt additional experimentation to confirm that SytVII does indeed reside on an internal membrane*…

To address this, we carried out pHluorin-syt 7 quenching/de-quenching experiments; pHluorin-syt 1 was used as a control (please see also (Dean et al., 2012)). We confirmed that both syt 7 and 1 are largely targeted to internal membranes, but that a significant fraction of each protein is also present in the plasma membrane (revised Figure 3—figure supplement 3).

*… and provide further experimentation to address how SytVII could specifically influence recovery*.

Our current study confirmed that loss of syt 7 has no effect on mEPSCs, or single action potential evoked synchronous and asynchronous release, and we further investigated the function of syt 7 on SV priming (hypertonic sucrose), recycling (pHluorin) and replenishment (current work). We found that only SV replenishment was impaired in syt 7 KO neurons, establishing specificity for the reported phenotype. Moreover, this phenotype has not been described for any other member of the syt family, including syt 1, 2, 4, 9, 10 ([5]; Dean et al., 2012; Dean et al., 2009; [24]; [27]; [33]; [42]; [48]), further establishing the novelty and specificity of our findings.

To further address the referees’ concerns, we compared the abilities of syt 1, 2, 4, 7, 9 and 10 to bind to CaM (again, since KO phenotypes have been described for these isoforms). Strikingly, only syt 7 exhibited robust Ca^2+^-dependent CaM binding activity (revised Figure 7); these data establish the highly specific nature of the Ca^2+^•syt 7•CaM interaction. Since CaM has emerged as a key factor for replenishment in a variety of earlier studies (Hosoi et al., 2007; Lipstein et al., 2013; Sakaba and Neher, 2001), this biochemical interaction provides a physical link between the physiological function of syt 7 and a known protein for SV replenishment, CaM. Indeed, we show that a CaM antagonist phenocopies loss of syt 7, suggesting that these proteins operate together to regulate replenishment in neurons.

*The interation with calmodulin is interesting, but it remains a protein interaction without a clearly understood relevance to vesicle recovery*.

It has been rigorously established that CaM regulates SV replenishment (17; 23; 37) and yes, at present it is unknown as to how CaM exerts its effects in this process: no earlier paper has worked-out a molecular mechanism. Our study is novel as we have uncovered a novel CaM binding protein, syt 7, as a key regulator of SV replenishment. So, at present, three molecules have been identified: CaM, munc13-1, and now, syt 7, in the replenishment pathway. Interestingly, both syt 7 and munc13-1 bind to CaM, so CaM perhaps serves as a central ‘hub’ for replenishment, and this issue will prompt further study. For example, we are keen to determine whether CaM can bind to both syt 7 and munc13-1 simultaneously. So, by identifying the key components, we can begin to fit them together into a molecular mechanism. It is likely that a model will await identification of additional components, but the current state-of-the-art is to first identify these components, as we have now done for syt 7 (and as Lipstein et al. (2013) have done by showing CaM works together with munc13-1 to regulate replenishment). We now state, in the “Discussion” section of the revised manuscript, that:

“In molecular terms, the emerging view is that CaM regulates SV replenishment via interactions with both Munc13-1 (Lipstein et al., 2013) and syt 7. Future studies are needed to determine whether CaM can physically interact with both proteins simultaneously, or whether CaM forms mutually exclusive complexes in the replenishment pathway.”

*3) The work provides no mention of the number of animals or samples used for any of the experiments, in either the Methods or the figure legends. The legend to*
Figure 1
*mentions three independent trials, but it is unclear what that means (animals, slices, cells), or its relationship to the statistics. This is a serious deficiency and a low n could greatly reduce confidence in this entire study. Similarly, the imaging in*
Figure 5
*presents unnormalized changes in fluorescence, and it is very hard to evaluate this either-it would be particularly difficult to combine data like this from multiple independent experiments, again raising concern about n*.

We apologize for the omission of key information, including the number of animals, statistical analysis etc.; this critical information has been added to each figure legend of the revised manuscript.

Panel A in Figure 5 is a representative raw image showing that our batch of bafilomycin is active: SVs fail to reacidify in the presence of the drug. This is a well-established effect of this drug, and is commonly used; we simply included a trace to make sure the drug was effective (this control trace is not normally included in publications and could be removed to save space). In contrast, the statistical analysis in panels B and C, in the same figure, were calculated based on the normalized fluorescence changes from three independent experiments (three distinct litters of mice), six coverslips (two from each mouse), and 50 boutons per coverslip, which is ∼ 300 boutons in total. We have added this information in the revised manuscript.

*4) The authors demonstrate that the D/N mutations in the C2 domain do not rescue, consistent with the need for this tagmin to bind calcium for its function. However the authors never actually show that this protein is actually present at synapses when re-expressed as a mutant. This should be addressed experimentally*.

We thank the referees for this suggestion. We have provided the requested data in the revised manuscript (revised Figure 3—figure supplement 2); the sub-cellular distribution of the mutant was the same as for the transfected WT protein.